# Conformational dynamics and allosteric modulation of the SARS-CoV-2 spike

**Marco A Díaz-Salinas[1], Qi Li[2], Monir Ejemel[2], Leonid Yurkovetskiy[3], Jeremy Luban[3,4], Kuang Shen[3], Yang Wang[2], James B Munro[1,4]\***

[1]Department of Microbiology and Physiological Systems, University of Massachusetts Chan Medical School, Worcester, United States; [2]MassBiologics of the University of Massachusetts Chan Medical School, Boston, United States; [3]Program in Molecular Medicine, University of Massachusetts Chan Medical School, Worcester, United States; [4]Department of Biochemistry and Molecular Biotechnology, University of Massachusetts Chan Medical School, Worcester, United States

**Abstract** Severe acute respiratory syndrome coronavirus 2 (SARS-CoV-2) infects cells through binding to angiotensin-converting enzyme 2 (ACE2). This interaction is mediated by the receptor-binding domain (RBD) of the viral spike (S) glycoprotein. Structural and dynamic data have shown that S can adopt multiple conformations, which controls the exposure of the ACE2-binding site in the RBD. Here, using single-molecule Förster resonance energy transfer (smFRET) imaging, we report the effects of ACE2 and antibody binding on the conformational dynamics of S from the Wuhan-1 strain and in the presence of the D614G mutation. We find that D614G modulates the energetics of the RBD position in a manner similar to ACE2 binding. We also find that antibodies that target diverse epitopes, including those distal to the RBD, stabilize the RBD in a position competent for ACE2 binding. Parallel solution-based binding experiments using fluorescence correlation spectroscopy (FCS) indicate antibody-mediated enhancement of ACE2 binding. These findings inform on novel strategies for therapeutic antibody cocktails.

**\*For correspondence:** james.munro@umassmed.edu

## Editor's evaluation

Using single-molecule fluorescence imaging, the authors of this paper characterize a conformational change of the Spike protein from severe acute respiratory syndrome coronavirus 2 (SARS-CoV-2) that is important for the ability of the Spike protein to target its receptor on the surface of a host cell and facilitate viral entry into the cell. The authors characterize how interactions of the Spike protein with its receptor and with Spike protein-targeting neutralizing antibodies alter this conformational change. The results provide important insights into the mechanisms-of-action of several classes of neutralizing antibodies.

## Introduction

Severe acute respiratory syndrome coronavirus 2 (SARS-CoV-2) is the etiologic agent of the coronavirus disease 2019 (COVID-19) pandemic (*Zhou et al., 2020b*). Despite the existence of efficacious COVID-19 vaccines (*Fan et al., 2021*), urgent needs remain for preventative and therapeutic strategies to mitigate the emergence of new variants of concern (*Rana et al., 2021*).

To infect host cells, SARS-CoV-2 binds the cell receptor angiotensin-converting enzyme 2 (ACE2) through its envelope glycoprotein spike (S), which subsequently promotes membrane fusion and cell entry (*Hoffmann et al., 2020*; *Lan et al., 2020*; *Letko et al., 2020*; *Shang et al., 2020*; *Walls et al., 2020*; ; *Wang et al., 2020*; *Wrapp et al., 2020*; *Yan et al., 2020*; *Zhou et al., 2020b*). S is a trimer

**Figure 1.** Single-molecule Förster resonance energy transfer (smFRET) imaging of the conformational dynamics of the severe acute respiratory syndrome coronavirus 2 (SARS-CoV-2) S ectodomain. (**A**) (Left) SARS-CoV-2 SΔTM containing a single fluorescently labeled A4-tagged protomer within an otherwise untagged trimer was immobilized on a streptavidin-coated quartz microscope slide by way of a C-terminal 8x-His-tag and biotin-NiNTA. For clarity, only a monomer is depicted. Individual SΔTM trimers were visualized with prism-based TIRF microscopy using a 532 nm laser. Overlay of two S protomers with receptor-binding domains (RBD) in the 'up' (blue) and 'down' (green) conformations are shown with approximate positions of fluorophores indicated by green (LD550) and red (LD650) stars. (Right) Top view of the same S protomer overlay. The approximate distances between the sites of labeling are shown. Structures adapted from PDB 6VSB. (**B**) Domain organization of the SARS-CoV-2 SΔTM construct used for smFRET experiments, indicating the sites of A4 tag insertion. The S1 and S2 subunits are in blue and orange, respectively. Additional domains and features are as follows, ordered from N- to C-terminus: signal peptide, dark green; NTD, N-terminal domain; RBD and RBM, receptor-binding domain and motif (purple), respectively; SD1, subunit domain 1; SD2, subunit domain 2; SGAG, furin cleavage site mutation; FP, fusion peptide; HR1 and HR2, heptad repeat 1 and 2, respectively; PP, diproline mutations; T4 fibritin trimerization motif (foldon), magenta; TEV protease cleavage site, brown; 8x-His-tag, green. (Bottom) Amino acid sequence alignments indicating sites of A4 tag insertions in SΔTM. A4 peptide sequences (DSLDMLEW) are underlined. Fluorophores get attached to the serine amino acid within the A4 peptide.

The online version of this article includes the following source data and figure supplement(s) for figure 1:

**Figure supplement 1.** Purification and labeling of severe acute respiratory syndrome coronavirus 2 (SARS-CoV-2) SΔTM and soluble angiotensin-converting enzyme 2 (ACE2).

**Figure supplement 1—source data 1.** Numeric chromatography data from purification of SΔTM and angiotensin-converting enzyme 2 (ACE2), and original Western blot images.

**Figure supplement 2.** Characterization of SΔTM containing A4 peptide tags for site-specific fluorophore attachment.

**Figure supplement 2—source data 1.** Numeric angiotensin-converting enzyme 2 (ACE2)-bound fraction data.

of heterodimers, with each protomer consisting of S1 and S2 subunits (*Figure 1*). S1 contains the receptor-binding domain (RBD), which includes the ACE2 receptor-binding motif (RBM). S2, which forms the spike stalk, undergoes a large-scale refolding during promotion of membrane fusion (*Cai et al., 2020*; *Tortorici and Veesler, 2019*; *Walls et al., 2017*; *Zhang et al., 2021b*). Structures of the soluble trimeric ectodomain of the SARS-CoV-2 S glycoprotein in two prefusion conformations have been reported (*Walls et al., 2020*; *Wrapp et al., 2020*; *Yurkovetskiy et al., 2020*). These distinct conformations demonstrate that the RBD of each protomer can independently adopt either a 'down' (closed) or an 'up' (open) position, giving rise to asymmetric trimer configurations (*Figure 1A*). The RBM is occluded in the down conformation, suggesting that the RBD must transition to the up conformation to bind ACE2. Indeed, structures of S bound to ACE2 show the RBD in the up conformation (*Zhou et al., 2020a*). Structural data were corroborated by real-time analysis of the conformational dynamics of S through single-molecule Förster resonance energy transfer (smFRET) imaging (*Lu et al., 2020*).

These structural and biophysical data suggest that modulating the conformational equilibrium of the RBD of S might be a determinant of SARS-CoV-2 infectivity and neutralization sensitivity. By the summer of 2020, the SARS-CoV-2 S variant D614G (B.1 lineage) had supplanted the ancestral virus (strain Wuhan-1) worldwide, and structural analysis showed that D614G disrupts an interprotomer

contact (*Zhang et al., 2021a*). This disruption results in a shift in the RBD conformation toward the up position, which is competent for ACE2 binding, consistent with increases in in vitro virus-cell binding mediated by ACE2 and infectivity (*Korber and Fischer, 2020*; *Yurkovetskiy et al., 2020*). At the same time, the enhanced exposure of the RBM in the D614G variant led to increased sensitivity to neutralizing antibodies (*Weissman et al., 2020*). Furthermore, the RBD showed stabilization in the up position, as well as an intermediate conformation, upon treatment with a neutralizing S2 stalk-directed antibody (*Li et al., 2022a*; *Ullah et al., 2021*).

Here, we report on the conformational dynamics of SARS-CoV-2 S in the absence or presence of ligands visualized using an smFRET imaging assay (*Figure 1A*). Our results indicate that ACE2 binding is controlled by the conformational dynamics of the RBD, with ACE2 capturing the intrinsically accessible up conformation rather than inducing a conformational change. We find that antibodies that target diverse epitopes—including epitopes in the N-terminal domain (NTD) and in the S2 stalk, which are distal to the RBD—tend to shift the RBD equilibrium on the D614 spike toward the up conformation, enhancing ACE2 binding. The D614G spike existed in an equilibrium where the RBD favors the up conformation prior to antibody binding. Nonetheless, antibodies that target the S2 stalk further promoted the RBD-up conformation on the D614G spike. We thus observe long-range allosteric modulation of the RBD equilibrium, which in turn regulates exposure of the ACE2-binding site. Inducing exposure of key neutralizing epitopes with antibodies will inform the design of novel therapeutic cocktails (*Corti et al., 2021*; *Hurt and Wheatley, 2021*; *Sun and Ho, 2020*).

## Results

### Tagged SARS-CoV-2 S maintains a native conformation

With the aim of visualizing the conformational dynamics in real time of SARS-CoV-2 S, we developed an smFRET imaging assay. We specifically sought to probe the movement of the RBD between the up and down positions. To this end, guided by the available structural data (*Walls et al., 2020*; *Wrapp et al., 2020*; *Figure 1A*), we inserted the 8-amino-acid A4 peptide into the spike ectodomain (SΔTM) within loops located between the β7 and β8 strands in the NTD at position 161, and between helix α1

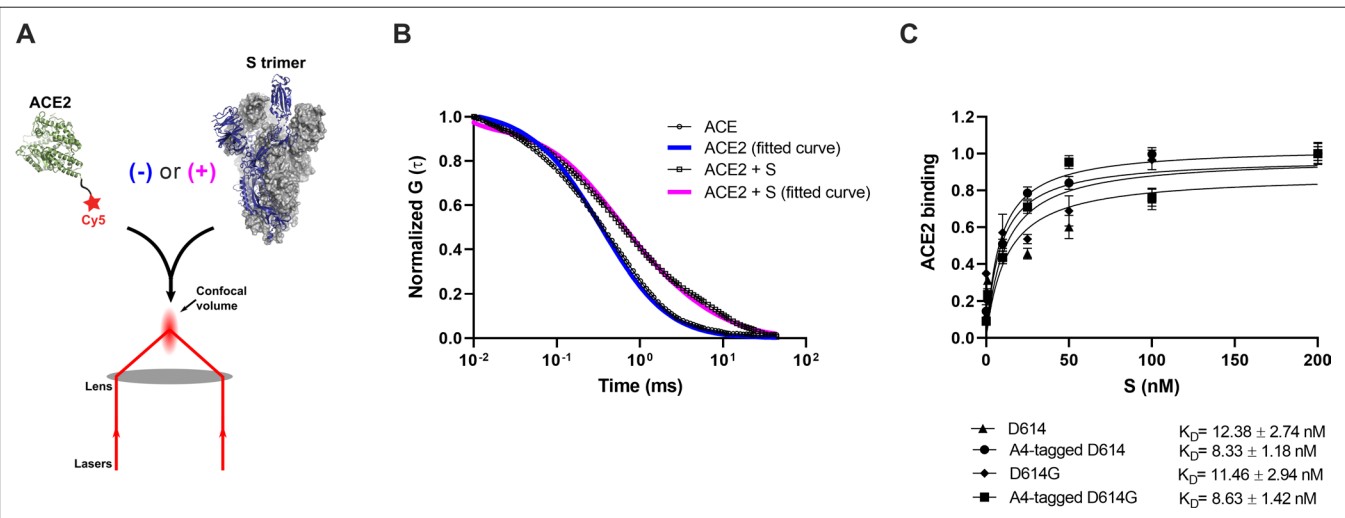

**Figure 2.** Verification of angiotensin-converting enzyme 2 (ACE2)-binding to A4-tagged SΔTM trimers using fluorescence correlation spectroscopy (FCS). (**A**) Cy5-labeled ACE2 was incubated in the absence or presence of untagged or A4-tagged SΔTM spikes. The diffusion of Cy5-ACE2 was evaluated by FCS using a 647 nm laser as indicated in Materials and methods. (**B**) Representative normalized autocorrelation curves for Cy5-ACE2 in the absence (circles) or presence (squares) of SΔTM, and the corresponding fits are shown in blue or magenta, respectively. The shift in the autocorrelation to longer timescales seen in the presence of SΔTM reflects the slower diffusion resulting from the larger size of the complex. (**C**) Cy5-ACE2 (100 nM) was incubated with different concentrations of the indicated SΔTM spikes and the resulting mixture was evaluated by FCS as described in Materials and methods. Dissociation constants ($K_D$) determined from fitting the titration are indicated for the different SΔTM constructs. Data are presented as the mean ± standard deviation from three independent measurements. Raw data is provided in *Figure 2—source data 1*.

The online version of this article includes the following source data for figure 2:

**Source data 1.** Numeric autocorrelation and bound angiotensin-converting enzyme 2 (ACE2) fraction data for panels B and C.

and strand β1 in the RBD at position 345 (*Figure 1B*). Fluorophores were then enzymatically attached to the A4 tags through incubation with AcpS (*Zhou et al., 2008*). This approach was chosen because it was previously used for conformational dynamics studies of S, as well as the spike proteins from HIV-1 and Ebola virus (*Alsahafi et al., 2019*; *Durham et al., 2020*; *Lu et al., 2020*; *Munro et al., 2014*). Structural analysis indicated an increase in the distance between the attachment sites of LD550 and LD650 fluorophores after the RBD transitions from the down to the up conformation, suggesting that this labeling strategy would allow us to visualize this dynamic event (*Figure 1A*; *Lu et al., 2020*).

Before proceeding to smFRET imaging, we first sought to validate the structure and antigenicity of the 161/345A4-tagged SΔTM trimer. Purified homo-trimers with either D614 or D614G were validated through two different approaches: (1) evaluation of their binding to ACE2 and (2) evaluation of their antigenic characteristics compared with untagged SΔTM. We developed a fluorescence correlation spectroscopy (FCS) assay to evaluate ACE2-binding to A4-tagged SΔTM trimers in solution (*Figure 2A*). For this assay, purified ACE2 was conjugated to the Cy5 fluorophore (*Figure 1—figure supplement 1*). Cy5-ACE2 was incubated with varying concentrations of either tagged or untagged SΔTM and the timescale of diffusion was measured using FCS. The FCS data were fit to a model of two diffusing species (*Lakowicz, 2006*; *Figure 2B*). Fitting led to determination of diffusion times for unbound ACE2 ($\tau_{free}$ = 0.48±0.02 ms), and ACE2 bound to SΔTM D614 ($\tau_{D614\text{-}bound}$ = 4.53±0.11 ms) or to the D614G variant ($\tau_{D614G\text{-}bound}$ = 4.32±0.15 ms). As expected, the diffusion times for the SΔTM-ACE2 complex were higher than for unbound ACE2, consistent with the formation of a larger complex with slower diffusion. Moreover, FCS experiments allowed us to calculate dissociation constants ($K_D$) for ACE2 binding to untagged and A4-tagged SΔTM proteins in solution (*Figure 2C*), which were approximately 12.4 ± 2.7 and 8.3 ± 1.2 nM, respectively, in rough agreement with values obtained through surface-based assays (*Walls et al., 2020*; *Yurkovetskiy et al., 2020*). The antigenicity of A4-tagged SΔTM homo-trimers was evaluated through an ELISA using the RBD-targeting antibodies MAb362 (both IgG$_1$ and IgA$_1$) (*Ejemel et al., 2020*), REGN10987 (*Hansen et al., 2020*), S309 (*Pinto et al., 2020*), and CR3022 (*Yuan et al., 2020*); NTD-targeting antibody 4A8 (*Chi et al., 2020*); and the stalk-targeting antibodies 1A9 (*Zheng et al., 2020*) and 2G12 (*Williams et al., 2021*). A4-tagged SΔTM homo-trimers maintained more than 50% of antibody binding compared to untagged SΔTM (*Figure 1—figure supplement 2*), with MAb362-IgG1 and 4A8 showing no statistically significant loss of binding. Taken together, these results suggest that double 161/345 A4-labeled SΔTM trimers maintain native functionality during ACE2 binding and near-native antigenic properties.

## Effects of ACE2 on the SARS-CoV-2 S RBD conformational equilibrium

To monitor the conformational dynamics of SΔTM D614 and D614G, we purified SΔTM hetero-trimers, formed by co-transfection of 161/345A4-tagged and untagged SΔTM plasmids at a 1:2 ratio. This ensured that on average the SΔTM trimers were comprised of one tagged protomer and two untagged protomers. The SΔTM hetero-trimers were then labeled with equimolar concentrations of LD550 and LD650 fluorophores. The labeled trimers were then incubated in the absence or presence of ACE2 before immobilization on a quartz microscope slide and imaging with TIRF microscopy. smFRET trajectories acquired from individual unbound SΔTM D614 molecules showed transitions between high (~0.65) and low (~0.35) FRET states, suggestive of the down and up RBD positions, respectively (*Figure 3A–B*). Hidden Markov modeling (HMM) confirmed that a kinetic model with two non-zero FRET states, and a 0-FRET state corresponding to photobleaching, was sufficient to describe the dynamics observed in the smFRET trajectories (*Figure 3—figure supplement 1*). Consistent with SΔTM D614 preferring the down conformation, HMM analysis indicated 60% ± 3% occupancy in the high-FRET state and 40% ± 3% occupancy in the low-FRET state (*Figure 3C*). The same FRET states were detected after incubation with ACE2, but the conformational equilibrium shifted to 34% ± 3% in the high-FRET state and 66% ± 3% occupancy in the low-FRET state (*Figure 3B–C*). This result is consistent with ACE2 promoting the RBD-up conformation. HMM analysis also indicated a reduction in the overall dynamics upon ACE2 binding, as indicated by the transition density plots (TDPs; *Figure 3D*), which display the relative frequencies of transitions between the high- and low-FRET states. The rates of transition were determined through maximum likelihood optimization of the three-state kinetic model. This analysis indicated that transition from the RBD-down (high FRET) to the RBD-up (low FRET) conformation occurred at $k_{up}$ = 2.6 ± 0.2 s$^{-1}$, whereas the RBD-up to RBD-down transition occurred at $k_{down}$ = 3.8 ± 0.2 s$^{-1}$. ACE2 binding had minimal effect on the RBD-down

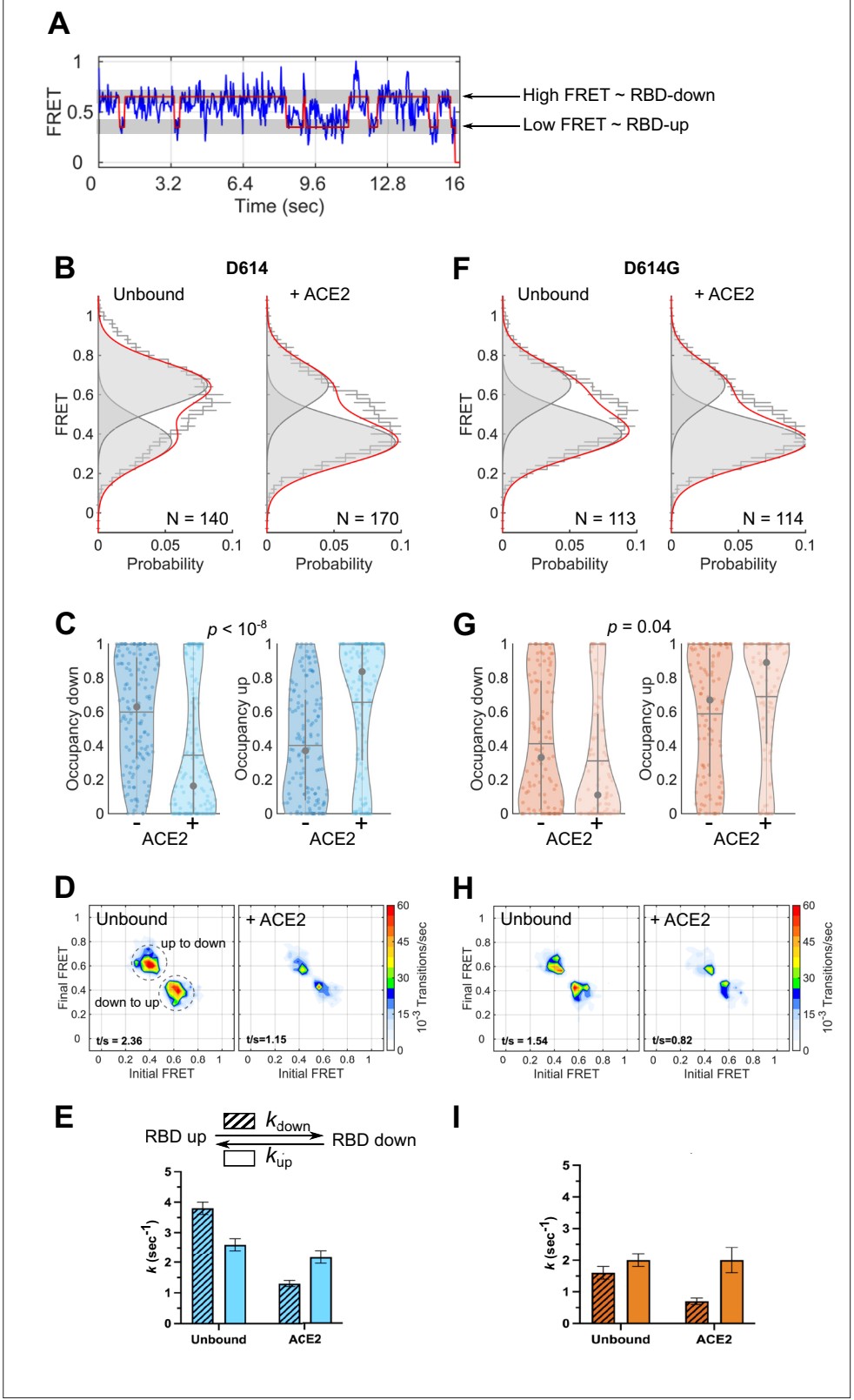

**Figure 3.** Angiotensin-converting enzyme 2 (ACE2)-binding modulates the receptor-binding domain (RBD) conformation of SΔTM D614 and D614G. (**A**) Representative single-molecule Förster resonance energy transfer (smFRET) trajectory acquired from an individual SΔTM trimer (blue). Idealization resulting from Hidden Markov modeling (HMM) analysis is overlaid (red). The high-FRET (0.65) and low-FRET (0.35) states correspond to the

*Figure 3 continued on next page*

*Figure 3 continued*

RBD-down and RBD-up conformations, respectively, as indicated. Bulk fluorescence lifetime and anisotropy measurements supported the interpretation of changes in FRET as arising due to conformational transitions that reposition the fluorophores and are presented in *Table 1*. (**B**) (Left) FRET histogram for unbound SΔTM D614 overlaid with the sum of two Gaussian distributions centered at 0.65 and 0.35 FRET (sum, red; single distributions, gray) generated from the results of HMM analysis. FRET histograms are presented as the mean ± standard error determined from three technical replicates. The total number of smFRET traces used in the HMM analysis is shown (**N**). (Right) The same data for the ACE2-bound SΔTM D614 spike. (**C**) Violin plots indicating the distribution of occupancies in the 0.65-FRET (RBD-down) and 0.35-FRET (RBD-up) states seen for the smFRET traces analyzed. For each plot the gray circles and horizontal lines indicate the median and mean occupancy, respectively. The vertical gray lines extend to the 25th and 75th quantiles. The statistical significance of the differences in occupancies seen for the unbound and ACE2-bound SΔTM D614 trimers was evaluated with a one-way ANOVA (p-value is indicated). (**D**) Transition density plots (TDPs) for (left) unbound and (right) ACE2-bound SΔTM D614 indicating the frequency of observed FRET transitions determined through HMM analysis. The assignment of the two transitions is indicated on the left-hand TDP. (**E**) (Top) Kinetic scheme defining the rates of transition between FRET states. (Bottom) Rates of transition for unbound and ACE2-bound SΔTM D614 determined from HMM analysis of the individual smFRET traces. Rate constants are presented as the mean ± standard error determined from the same populations of smFRET traces used to construct the FRET histograms. (**F**) FRET histograms for the unbound and ACE2-bound SΔTM D614G spike, displayed as in (**B**). (**G**) Violin plots indicating FRET state occupancies for the SΔTM D614G spike, displayed as in (**C**). (**H**) TDPs for the SΔTM D614G spike, displayed as in (**D**). (**I**) Rate constants for the unbound and ACE2-bound SΔTM D614G spike, displayed as in (**E**). Numeric data are provided in *Figure 3—source data 1*.

The online version of this article includes the following source data and figure supplement(s) for figure 3:

**Source data 1.** Matlab figure files containing numeric data for Förster resonance energy transfer (FRET) histograms, violin plots, and transition density plots (TDPs), and numeric kinetics data.

**Figure supplement 1.** Selection of a model for analysis of single-molecule Förster resonance energy transfer (smFRET) trajectories.

to the RBD-up transition ($k_{up}$ = 2.2 ± 0.2 s$^{-1}$) but reduced the RBD-up to RBD-down transition to $k_{down}$ = 1.3 ± 0.1 s$^{-1}$ (*Figure 3E*). This analysis thus specified that the effect of ACE2 binding is to capture and stabilize the RBD-up conformation and reduce transitions to the RBD-down conformation. ACE2 binding does not significantly affect the stability of the down conformation, nor induce transitions to the up conformation.

We next sought to determine the effect of the D614G mutation on the conformational dynamics of SΔTM. We observed the same two FRET states for SΔTM D614G as for the ancestral D614 spike (*Figure 3F*). However, the unbound SΔTM D614G displayed greater occupancy in the RBD-up conformation (59% ± 3%), and the overall level of dynamics was reduced as compared to D614 (*Figure 3G–H*). The rate constants, $k_{up}$ and $k_{down}$, were reduced to 2.0 ± 0.2 and 1.6 ± 0.2 s$^{-1}$, respectively (*Figure 3I*). ACE2 binding further increased the RBD-up occupancy to 69% ± 4% and reduced the overall level of dynamics shown in the TDPs (*Figure 3G–H*). As seen for D614, ACE2 binding had minimal effect on the rate of transition from the RBD-down to the RBD-up conformation ($k_{up}$ = 2.0 ± 0.4 s$^{-1}$) but reduced the rate of transition to the RBD-down conformation to $k_{down}$ = 0.7 ± 0.1 s$^{-1}$ (*Figure 3I*). Thus, consistent

**Table 1.** Fluorescence lifetime and anisotropy measurements.

Data are presented as the mean ± standard error determined from three technical replicates. ACE2, angiotensin-converting enzyme 2; ND, not determined.

|  | Lifetime (ns) | Anisotropy |
|---|---|---|
| LD550 | 1.167 ± 0.004 | 0.143 ± 0.002 |
| SΔTM-LD550 | 1.463 ± 0.006 | 0.101 ± 0.003 |
| SΔTM-LD550 + ACE2 | 1.523 ± 0.008 | 0.093 ± 0.004 |
| LD650 | ND | 0.140 ± 0.001 |
| SΔTM-LD650 | ND | 0.258 ± 0.004 |
| SΔTM-LD650 + ACE2 | ND | 0.247 ± 0.003 |

with structural studies (*Yurkovetskiy et al., 2020*; *Zhang et al., 2021a*), the D614G mutation shifted the conformational equilibrium in favor of the RBD-up conformation. Also, here again, ACE2 binding stabilized the RBD-up conformation without affecting the energetics of the RBD-down conformation.

### RBD-targeting antibodies promote the RBD-up conformation of S D614

Numerous neutralizing monoclonal antibodies (mAbs) targeting SARS-CoV-2 S have been identified (*Li et al., 2022b*). However, their mechanisms of action have only been partially described, especially for mAbs that target epitopes outside of the RBD. We first sought to use our smFRET imaging approach to explore the effect of RBD-directed mAbs on SΔTM dynamics for both the D614 and D614G variants. We chose neutralizing RBD-directed mAbs from different classified groups according to the epitope targeted (*Gavor et al., 2020*): (1) MAb362 (isoforms IgG$_1$ and IgA$_1$) that directly targets the RBM (*Ejemel et al., 2020*); (2) REGN10987, which binds an epitope located on the side of the RBD, blocking ACE2 binding without directly interacting with the RBM (*Hansen et al., 2020*); (3) and S309 and CR3022 that bind the RBD but do not compete with ACE2 binding (*Pinto et al., 2020*; *Yuan*

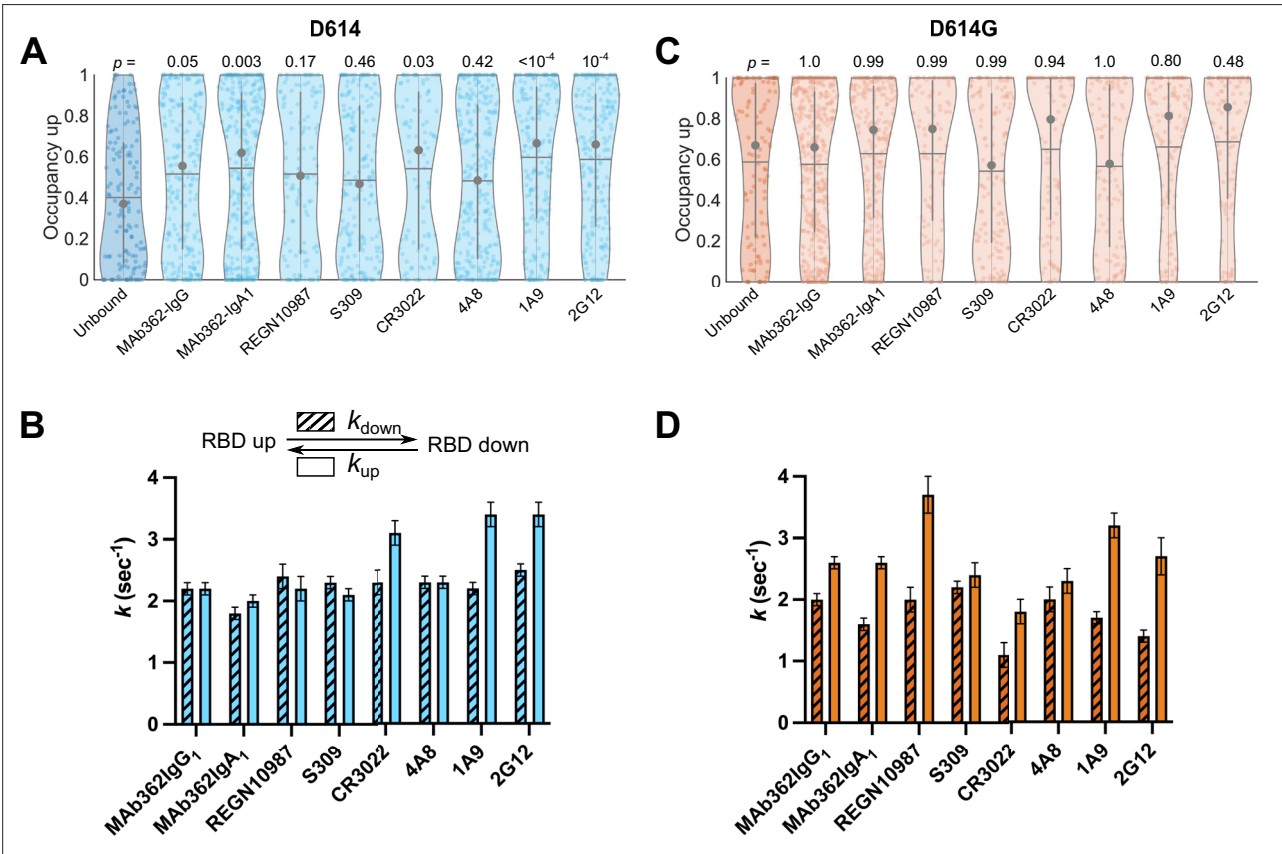

**Figure 4.** Antibodies directly and allosterically modulate SΔTM receptor-binding domain (RBD) conformation. (**A**) The RBD-up conformation occupancy (low-Förster resonance energy transfer [FRET] state) determined through Hidden Markov modeling (HMM) analysis for SΔTM D614 in the absence or presence of the indicated monoclonal antibodies (mAbs). Occupancy data are presented as violin plots as in *Figure 3*. The indicated p-values were determined by comparing mAb-bound to unbound SΔTM through one-way ANOVA. (**B**) (Top) Kinetic scheme defining the rates of transition between RBD-up and -down conformations. (Bottom) Rates of transition for SΔTM D614 in the presence of mAbs determined through HMM analysis of the single-molecule FRET (smFRET) traces. Rate constants are presented as mean ± standard error determined from three technical replicates. (**C**) RBD-up conformation occupancy data for SΔTM D614G displayed as in (**A**). (**D**) Kinetic data for SΔTM D614G displayed as in (**B**). Corresponding FRET histograms for each mAb-bound SΔTM trimer is shown in *Figure 4—figure supplement 1*. Data are shown numerically in *Tables 2 and 3* and provided in *Figure 4—source data 1*.

The online version of this article includes the following source data and figure supplement(s) for figure 4:

**Source data 1.** Matlab figure files containing numeric data for violin plots and numeric kinetics data.

**Figure supplement 1.** Förster resonance energy transfer (FRET) histograms for SΔTM in the presence of monoclonal antibodies (mAbs).

**Figure supplement 1—source data 1.** Matlab figure files contains numeric Förster resonance energy transfer (FRET) histogram data.

*et al., 2020*). Imaging SΔTM pre-incubated with each of the above mAbs revealed a predominant low-FRET state associated with the RBD-up conformation (*Figure 4—figure supplement 1*). In all cases, the mAbs stabilized the RBD-up conformation of SΔTM D614 as compared to the unbound spike, although none to the extent seen for ACE2 (*Figure 4A*); and the effect of REGN10987 and S309 did not reach statistical significance. As observed during ACE2 binding, the mAbs generally induced a larger effect on the rate of transition from the RBD-up to the RBD-down conformation ($k_{down}$), with a minor effect on the reverse transition ($k_{up}$; *Figure 4B*). In contrast, none of the mAbs stabilized the RBD-up conformation for SΔTM D614G to a significant extent (*Figure 4C–D*), suggesting that the effect of the D614G mutation is sufficient to enable mAb binding without further conformational changes.

## NTD- and Stalk-targeting mAbs allosterically modulate the RBD position

Several mAbs have been identified that target epitopes outside of the RBD, some of which bind the NTD and are potently neutralizing (*Cerutti et al., 2021*; *Chi et al., 2020*; *McCallum et al., 2021b*; *Suryadevara et al., 2021*). We therefore explored the conformational dynamics of both SΔTM D614 and D614G pre-treated with the NTD-targeting mAb 4A8 (*Chi et al., 2020*), and with the S2 stalk-directed mAbs 1A9 (*Zheng et al., 2020*) and 2G12 (*Williams et al., 2021*). 4A8 treatment of SΔTM D614 stabilized the low-FRET state to a comparable extent as seen for the RBD-targeted mAbs, although the effect did not reach statistical significance (*Figure 4A*). The change in transition rates also followed a similar trend as seen for RBD-targeted mAbs with the RBD-up to RBD-down ($k_{down}$) transition being reduced, with a minor effect on the reverse transition ($k_{up}$; *Figure 4B*). The stalk-directed mAbs 1A9 and 2G12 significantly stabilized the RBD-up conformation (*Figure 4A*), although kinetic analysis revealed a modulation of the dynamics that was distinct from the S1-targeted mAbs. Here, in both cases $k_{down}$ was reduced, while $k_{up}$ was increased (*Figure 4B*). For SΔTM D614G, 4A8 had only a minor effect on RBD-up occupancy or kinetics, again suggesting that the mAb binds without affecting the conformational equilibrium (*Figure 4C–D*). However, the stalk-targeting 1A9 and 2G12 mAbs stabilized RBD-up, though not to a statistically significant extent, and induced increases in $k_{up}$

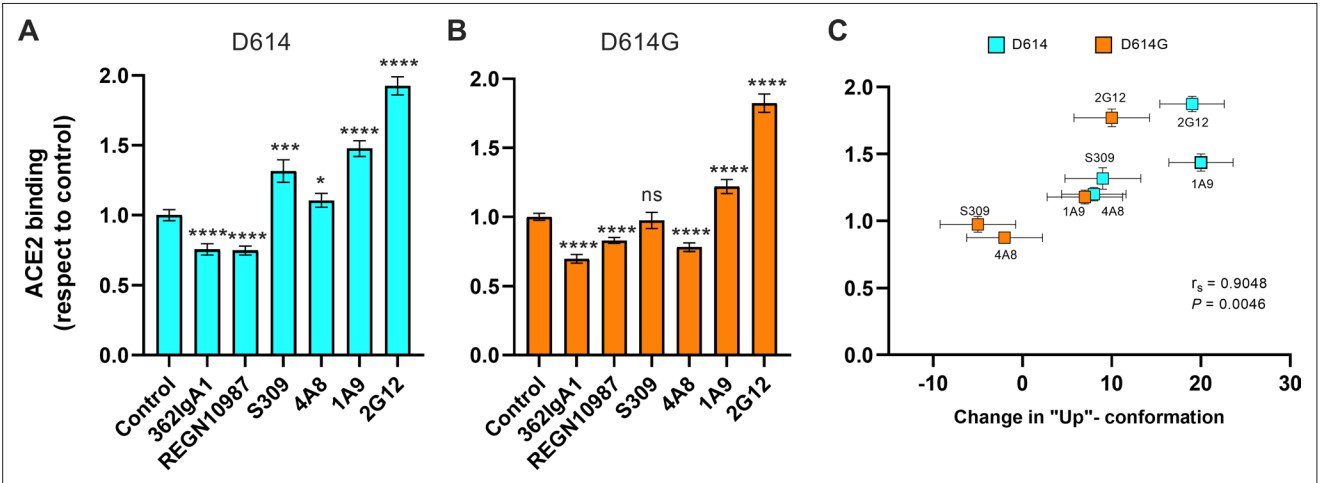

**Figure 5.** Allosteric modulation of the receptor-binding domain (RBD) position promotes angiotensin-converting enzyme 2 (ACE2) binding. (**A**) Binding of ACE2 by (**A**) SΔTM D614 or (**B**) D614G spikes pre-incubated with the indicated monoclonal antibodies (mAbs) was measured by fluorescence correlation spectroscopy (FCS) as described in Materials and methods. Data are presented as the average of two independent experiments, each consisting of 20–25 10 s acquisitions. Statistical significance was evaluated through a two-tailed, unpaired Mann-Whitney test as indicated in Materials and methods. p-Values < 0.05 were considered significant and significance values are indicated as *p < 0.05, **p < 0.01, ***p < 0.001, ****p < 0.0001. (**C**) The change in the RBD-up conformation of SΔTM spikes pre-incubated with the indicated mAbs exhibited a positive correlation with the binding of ACE2 determined through FCS. Statistical significance (p = 0.0046) was found when Spearman test was performed with the 95% level of confidence (α = 0.05). Raw data are provided in *Figure 5—source data 1*.

The online version of this article includes the following source data for figure 5:

**Source data 1.** Numeric angiotensin-converting enzyme 2 (ACE2)-bound fraction data, and numeric change in receptor-binding domain (RBD)-up conformation data.

(*Figure 4C–D*). These data indicate that the S2 stalk-targeting mAbs studied here allosterically induce transition of the RDB to the up conformation on both the D614 and D614G spikes. In contrast, the RBD- and NTD-targeting mAbs studied here capture the up conformation without actively inducing a conformational change, similar to the effects of ACE2 on the RBD conformation.

## Stalk-targeting mAbs allosterically enhance ACE2 binding

We next asked if stabilization of the RBD-up conformation by NTD- and stalk-targeted mAbs would increase ACE2 binding. We therefore applied our FCS assay for ACE2 binding after pre-treating SΔTM D614 or D614G with mAbs. MAb362IgA$_1$ and REGN10987 mAbs were used as controls because of their documented ACE2-blocking properties (*Ejemel et al., 2020*; *Hansen et al., 2020*). Incubation of SΔTM D614 or D614G with MAb362IgA$_1$ or REGN10987 resulted in statistically significant reductions in ACE2 binding that are consistent with previous reports at comparable concentrations (*Ejemel et al., 2020*; *Hansen et al., 2020*; *Figure 5A, B*). Overall, mAbs that stabilized the RBD-up conformation without blocking the ACE2-binding site tended to promote ACE2 binding (*Figure 5*). Calculation of the Spearman coefficient indicated a very strong correlation ($r_s$ = 0.9048) between ACE2 binding and modulation of the SΔTM RBD conformational equilibrium across all the mAbs under consideration (*Figure 5C*). S309 and 4A8 provided a slight enhancement of ACE2 binding to SΔTM D614, consistent with their modest impacts on RBD conformation. In contrast, S309 had no significant effect on ACE2 binding to SΔTM D614G, and 4A8 had a slight inhibition of ACE2 binding, again consistent with their modulation of RBD conformation. Of note, the stalk-targeting 1A9 and 2G12 mAbs induced the greatest enhancement of ACE2 binding to SΔTM D614 and D614G, consistent with their allosteric modulation of RBD conformation.

## Discussion

Time-resolved analysis of viral spike protein conformation at single-molecule resolution complements structural studies by specifying the effects of ligand binding on the energetics of conformational dynamics. These analyses provide mechanistic insights unattainable from structures and bulk functional data alone. Here, we have developed and applied an smFRET imaging approach to monitor conformational dynamics of SARS-CoV-2 S from the ancestral Wuhan-1 strain with D614 and the D614G variant (B.1 lineage) during engagement with the ACE2 receptor and mAbs. Our analysis of S conformational dynamics shows that ACE2 stabilizes the RBD in the up conformation, which, in agreement with structural data, is a conformation that pre-exists prior to ACE2 binding (*Walls et al., 2020*; *Wrapp et al., 2020*). Determination of the kinetics of conformational changes through HMM analysis indicated that ACE2 binding does not affect the rate of transition to the RBD-up conformation. Instead, ACE2 captures the RBD-up conformation and reduces the rate of transition to the RBD-down conformation. This can be explained by a thermodynamic stabilization of the RBD-up conformation

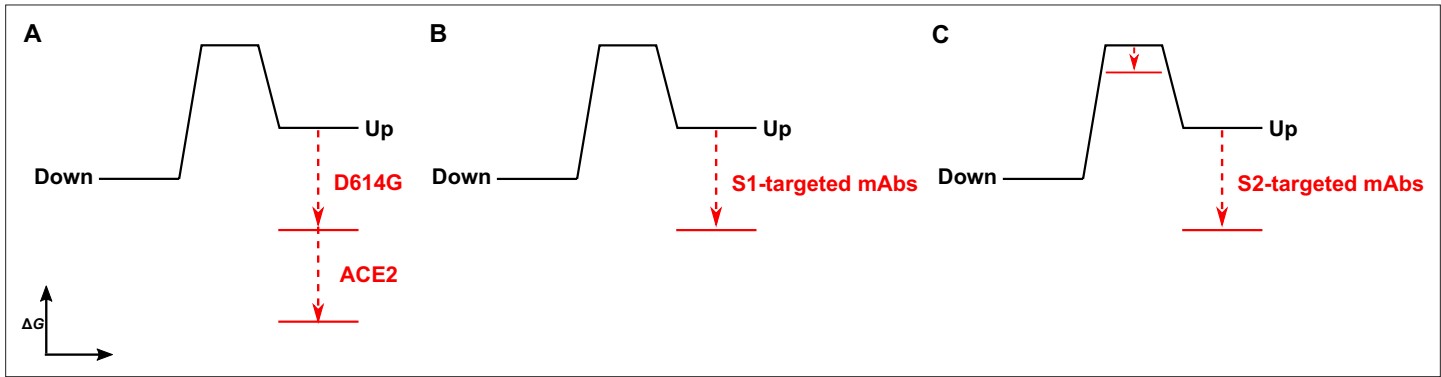

**Figure 6.** The D614G mutation and ligands modulate the S energetic landscape. (**A**) The D614G mutation and angiotensin-converting enzyme 2 (ACE2) have additive effects on the thermodynamic stabilization of the receptor-binding domain (RBD)-up conformation. (**B**) The predominant effect of monoclonal antibodies (mAbs) that target the S1 domain, either the RBD (MAb362, REGN10987, S309, CR3022) or N-terminal domain (NTD) (4A8), is to stabilize the RBD-up conformation. (**C**) mAbs that target the S2 domain have a more complex allosteric effect, resulting in stabilization of the RBD-up conformation coupled to reduction in the activation energy for transition from the RBD-down to the -up conformation.

without affecting the energetics of the down conformation (*Figure 6A*). This analysis of S dynamics specifies that ACE2 binding to S does not induce a conformational change in S, but rather occurs through the capture of a pre-existing conformation.

As ACE2 binding is an essential step during SARS-CoV-2 entry, our interpretation implicates the intrinsic dynamics of S in controlling the rate or efficiency of membrane fusion during virus entry. Current models of coronaviral membrane fusion depict the RBD-up conformation as an intermediate state that is on-pathway to the post-fusion S conformation (*Pallesen et al., 2017*; *Walls et al., 2019*). Accordingly, factors that stabilize the RBD-up conformation would likely increase the rate of membrane fusion. Our data demonstrate that the D614G mutation stabilized the RBD-up conformation, consistent with previous reports, which likely relates to how the mutation enhances infectivity (*Yurkovetskiy et al., 2020*; *Zhang et al., 2021a*). Previous studies have shown that D614G does not increase the rate of ACE2 binding to S (*Yurkovetskiy et al., 2020*; *Zhang et al., 2021a*), as might be expected for a conformational capture-binding mechanism. This may indicate that further rearrangements in the RBD, perhaps localized in the RBM, are necessary to fully engage ACE2 beyond transition to the up conformation. Analysis of the dynamics of the unbound D614G variant showed an overall reduction in dynamics as compared to D614, consistent with the increased thermostability of the S trimer with the D614G mutation (*Zhang et al., 2021a*). Like ACE2 binding to the D614 spike, the predominant effect of the D614G mutation was the reduction of the rate of transition to the down conformation. This observation indicates an increase in the activation energy, which is mainly explained by an increase in thermodynamic stability of the RBD-up conformation (*Figure 6A*). ACE2 binding to S D614G had an additive effect on the RBD position, pushing the equilibrium further toward the up conformation than either ACE2 binding or the D614G mutation did independently. Thus, the D614G mutation permits further stabilization of an intermediate conformation captured by ACE2 binding. Here again, ACE2 binding functioned by specifically increasing the thermodynamic stability of the RBD-up conformation. Residue D614 is distal to the RBD and forms a salt bridge with K854 in the fusion-peptide proximal region, which is lost with the D614G mutation (*Xiong et al., 2020*; *Zhang et al., 2021a*; *Zhou et al., 2020a*). Our analysis shows that the D614-K854 electrostatic interaction had a destabilizing effect on the thermodynamics of the RBD-up conformation. The similar impacts of D614G and ACE2 binding on the S energetic landscape imply that the mutation provides a fitness advantage by mimicking the effects of receptor binding. Such a long-range allosteric connection between the RBD and the region surrounding the fusion peptide has been reported for influenza hemagglutinin and the Ebola virus envelope glycoprotein (*Das et al., 2018*; *Das et al., 2020*; *Yewdell et al., 1993*). This suggests a common mechanistic connection between receptor binding and triggering movement of the fusion peptide (or fusion loop) among class-I viral fusion proteins.

We find that mAbs that target S1 of the D614 spike, including the RBD and NTD, have a similar impact on conformational dynamics as ACE2, with the predominant effect being the reduction in the rate of transition to the down conformation. The overall minimal effect on the rate of transition to the up conformation is again consistent with thermodynamic stabilization and the capture of a pre-existing S conformation (*Figure 6B*). mAb S309 had a notably modest effect on the stability of the up conformation, consistent with structural data demonstrating that it binds to the RBD in either the up or down positions (*Pinto et al., 2020*). Overall, RBD- and NTD-targeting mAbs had minimal effect on the conformation of the D614G spike. Of note, mAb S309 modestly destabilized the RBD-up conformation. As S309 does not prevent ACE2 binding, its mechanism of broad neutralization remains unclear (*McCallum et al., 2021a*; *Pinto et al., 2020*). However, its modulation of the RBD position likely plays some role and may impact downstream conformational changes related to membrane fusion.

Our kinetic analyses have shown that the S1-targeted ligands considered here capture the RBD-up conformation. In contrast, the S2-targeted mAbs considered here induce conformational changes in the RBD by reducing the activation energy for transition into the up conformation, while also stabilizing the up conformation (*Figure 6C*). Cryo-electron tomography of SARS-CoV-2 virions has revealed the presence of three flexible hinges within the S stalk: the hip, knee, and ankle. These hinges connect the head, and the upper and lower legs of S, and confer flexibility of the spikes (*Ke et al., 2020*; *Turoňová et al., 2020*). Our smFRET data demonstrate that stalk-targeted mAbs 1A9 and 2G12 allosterically modulate the position of the RBD, enhancing ACE2 binding. mAb 1A9, which neutralizes SARS-CoV, binds an epitope on S in the upper leg of the stalk near the hip and upstream of the heptad repeat

two helix (*Lip et al., 2006*). High sequence conservation in the 1A9 epitope suggests a similar binding site in SARS-CoV-2 S and mode of action in preventing viral membrane fusion (*Zheng et al., 2020*). The stalk epitope recognized by mAb 2G12, which does not neutralize SARS-CoV-2, is located near the hip and is comprised entirely of glycans (*Williams et al., 2021*). Taken together, our data on 1A9 and 2G12 implicate the hip hinge as a critical center for allosteric control of the RBD position. Further support for the existence of allosteric centers in S2 came from other smFRET analyses of mAb CV3-25 (*Jennewein et al., 2021*), which binds an epitope in the upper leg of the stalk near the knee (*Li et al., 2022a*). CV3-25 was also found to promote the RBD-up conformation (*Li et al., 2022b*; *Ullah et al., 2021*). Further studies are necessary to determine whether mAbs that target the lower leg and ankle hinge also exert allosteric control of the RBD.

The use of therapeutic mAb cocktails is a promising strategy, which has been explored for the treatment and prevention of Ebola virus disease (*Misasi and Sullivan, 2021*). Similarly, enhancement of neutralization of SARS-CoV-2 was observed with the simultaneous use of S309 and S2E12 (*Starr and Czudnochowski, 2021*; *Tortorici et al., 2020*) which targets the RBM. This likely stems from the combined effect of S309 on S conformation and blocking ACE2 binding by S2E12. Our results suggest that similar synergy in neutralization might come from the combination of 4A8 with RBM-directed mAbs. Indeed, human trials are underway evaluating mAb cocktails for COVID-19 treatment. But none of these have considered the simultaneous use of mAbs targeting the RBD and stalk of SARS-CoV-2 S (*Hurt and Wheatley, 2021*; *Li et al., 2022a*). The promotion of the RBD-up conformation, which exposes the ACE2-binding site, by NTD-directed mAbs like 4A8, or stalk-directed mAbs like 1A9 and 2G12, presents a strategy for enhancement of neutralization through combination therapies with RBM-directed mAbs. The results presented here suggest the potential for synergistic inhibition of virus entry and increased potency through the combination of mAbs that target diverse epitopes.

# Materials and methods

## Key resources table

| Reagent type (species) or resource | Designation | Source or reference | Identifiers | Additional information |
|---|---|---|---|---|
| Cell line (*Cricetulus griseus*) | ExpiCHO-S | Gibco, Thermo Fisher Scientific (Waltham, MA) | Cat. No. A29127 | |
| Cell line (*Homo sapiens*) | Expi293F | Gibco, Thermo Fisher Scientific (Waltham, MA) | Cat. No. A14527 | |
| Antibody | MAb362-IgG₁ (Human monoclonal) | PMID:32826914 | | ELISA (600 nM), FCS (600 nM), smFRET (see Materials and methods section 'smFRET imaging and data analysis'). |
| Antibody | MAb362-IgA₁ (Human monoclonal) | PMID:32826914 | | ELISA (600 nM), FCS (600 nM), smFRET (see Materials and methods section 'smFRET imaging and data analysis'). |
| Antibody | REGN10987 (Mouse monoclonal) | This work and PMID:32540901 | | See Materials and methods section 'Antibodies'. ELISA (600 nM), FCS (600 nM), smFRET (see Materials and methods section 'smFRET imaging and data analysis'). |
| Antibody | S309 (Human monoclonal) | This work and PMID:32422645 | | See Materials and methods section 'Antibodies'. ELISA (600 nM), FCS (600 nM), smFRET (see Materials and methods section 'smFRET imaging and data analysis'). |
| Antibody | CR3022 (Human monoclonal) | This work and PMID:16796401 | | See Materials and methods section 'Antibodies'. ELISA (600 nM), FCS (600 nM), smFRET (see Materials and methods section 'smFRET imaging and data analysis'). |

*Continued on next page*

*Continued*

| Reagent type (species) or resource | Designation | Source or reference | Identifiers | Additional information |
|---|---|---|---|---|
| Antibody | 2G12 (Human monoclonal) | This work | | See Materials and methods section 'Antibodies'. ELISA (600 nM), FCS (600 nM), smFRET (see Materials and methods section 'smFRET imaging and data analysis'). |
| Antibody | 4A8 (Human monoclonal) | BioVision (Milpitas, CA) | Cat. No. A2269-100 | ELISA (600 nM), FCS (600 nM), smFRET (see Materials and methods section 'smFRET imaging and data analysis'). |
| Antibody | 1A9 (Mouse monoclonal) | GeneTex (Irvine, CA) | Cat. No. GTX632604 | ELISA (600 nM), FCS (600 nM), SmFRET: see Materials and methods section 'smFRET imaging and data analysis', WB (1:2000). |
| Antibody | Anti-6x-His-tag (Rabbit polyclonal) | Invitrogen (Waltham, MA) | Cat. No. PA1-983B | WB (1:2000). |
| Antibody | HRP-conjugated anti-mouse IgG Fc (Rabbit polyclonal) | Invitrogen (Waltham, MA) | Cat. No. 31455 | ELISA (1:5000), WB (1:5000). |
| Antibody | HRP-conjugated anti-human IgG Fc (Goat polyclonal) | Invitrogen (Waltham, MA) | Cat. No. A18823 | ELISA (1:10,000), WB (1:10,000). |
| Antibody | HRP-conjugated anti-human kappa (Goat polyclonal) | SouthernBiotech (Birmingham, AL) | Cat. No. 2060–05 | ELISA (1:4000). |
| Antibody | HRP-conjugated anti-rabbit IgG (Goat polyclonal) | Abcam (Cambridge, UK) | ab205718 | WB (1:50,000). |
| Recombinant DNA reagent | pcDNA3.1_ REGN10987-heavy_chain | This work and PMID:32540901 | | See Materials and methods section 'Antibodies'. |
| Recombinant DNA reagent | pcDNA3.1_ REGN10987-light_chain | This work and PMID:32540901 | | See Materials and methods section 'Antibodies'. |
| Recombinant DNA reagent | pcDNA3.1_S309-heavy_chain | This work and PMID:32422645 | | See Materials and methods section 'Antibodies'. |
| Recombinant DNA reagent | pcDNA3.1_S309-light_chain | This work and PMID:32422645 | | See Materials and methods section 'Antibodies'. |
| Recombinant DNA reagent | pcDNA3.1_CR3022-heavy_chain | This work and PMID:16796401 | | See Materials and methods section 'Antibodies'. |
| Recombinant DNA reagent | pcDNA3.1_CR3022-light_chain | This work and PMID:16796401 | | See Materials and methods section 'Antibodies'. |
| Recombinant DNA reagent | Plasmid_2G12-heavy_chain | Peter D Kwong laboratory | | |
| Recombinant DNA reagent | Plasmid_2G12-light_chain | Peter D Kwong laboratory | | |
| Recombinant DNA reagent | pcDNA3.1 SARS-CoV-2 SΔTM (plasmid) | This work. GenScript (Piscataway, NJ) | | See Materials and methods section 'Plasmids and site-directed mutagenesis'. |
| Recombinant DNA reagent | pcDNA3.1 SARS-CoV-2 SΔTM 161/345A4 double-tagged (plasmid) | This work | | See Materials and methods section 'Plasmids and site-directed mutagenesis'. |
| Recombinant DNA reagent | pcDNA3.1 SARS-CoV-2 SΔTM D614G (plasmid) | This work | | See Materials and methods section 'Plasmids and site-directed mutagenesis'. |
| Recombinant DNA reagent | pcDNA3.1 SARS-CoV-2 SΔTM D614G 161/345A4 double-tagged (plasmid) | This work | | See Materials and methods section 'Plasmids and site-directed mutagenesis'. |

*Continued on next page*

*Continued*

| Reagent type (species) or resource | Designation | Source or reference | Identifiers | Additional information |
|---|---|---|---|---|
| Recombinant DNA reagent | pCAGGS-ACE2-his (plasmid) | PMID:32991842 | Addgene No. 158089 (Watertown, MA) | |
| Peptide, recombinant protein | SARS-CoV-2 SΔTM | This work | | See Materials and methods section 'Protein expression and purification'. |
| Peptide, recombinant protein | SARS-CoV-2 SΔTM 161/345A4 double-tagged | This work | | See Materials and methods section 'Protein expression and purification'. |
| Peptide, recombinant protein | SARS-CoV-2 SΔTM D614G | This work | | See Materials and methods section 'Protein expression and purification'. |
| Peptide, recombinant protein | SARS-CoV-2 SΔTM D614G 161/345A4 double-tagged | This work | | See Materials and methods section 'Protein expression and purification'. |
| Peptide, recombinant protein | ACE2 | This work | | See Materials and methods section 'Protein expression and purification'. |
| Peptide, recombinant protein | Acyl carrier protein synthase (AcpS) | PMID:31952255 | | |
| Commercial assay, kit | Q5 Site-Directed Mutagenesis Kit | New England Biolabs (Ipswich, MA) | Cat. No. E0554S | |
| Commercial assay, kit | Coomassie Plus (Bradford) Assay Kit | Thermo Fisher Scientific (Waltham, MA) | Cat. No. 23,236 | |
| Commercial assay, kit | SuperSignal West Pico PLUS Chemiluminescent Substrate | Thermo Scientific (Waltham, MA) | Cat. No. 34580 | |
| Commercial assay, kit | 1-Step Ultra TBM-ELISA | Thermo Scientific (Waltham, MA) | Cat. No. 34028 | |
| Chemical compound, drug | Cy5-conjugated n-hydroxysuccinimide ester | Cytiva (Marlborough, MA) | Cat. No. PA15100 | |
| Chemical compound, drug | Coenzyme A (CoA)-conjugated LD550 fluorophore | Lumidyne Technologies (New York, NY), | Cat. No. LD550-CS | |
| Chemical compound, drug | Coenzyme A (CoA)-conjugated LD650 fluorophore | Lumidyne Technologies (New York, NY), | Cat. No. LD650-CS | |
| Software, algorithm | Micromanager | PMID:25606571 micro-manager.org | v2.0 | |
| Software, algorithm | SPARTAN | https://www.scottcblanchardlab.com/software and PMID:26878382 | Version 3.7 | |
| Software, algorithm | Matlab (Mathworks, Natick, MA) | Mathworks (Natick, MA) | Version R2018b | |
| Software, algorithm | Maximum point likelihood algorithm | PMID:11023897 | | |
| Software, algorithm | GraphPad Prism | GraphPad Software (San Diego, CA) | Version 9.2.0 | |
| Software, algorithm | PyMOL software | The PyMOL Molecular Graphic System, Schrödinger Inc (New York, NY) | Version 2.0.7 | |
| Other | Pierce Protein G Agarose | Thermo Fisher Scientific (Waltham, MA) | Cat. No. 20398 | |
| Other | Ni-NTA Agarose | Invitrogen (Waltham, MA) | Cat. No. R901-15 | |
| Other | Superdex 200 Increase 10/300 GL column | GE Healthcare (Chicago, IL) | Cat. No. 28990944 | |
| Other | Gel Filtration Standard | Bio-Rad (Hercules, CA) | Cat. No. 1511901 | |
| Other | Synergy H1 microplate reader | BioTek (Winooski, VT) | | |
| Other | Typhoon 9,410 variable mode imager | GE Amersham Biosciences (Amersham, UK) | | |
| Other | CorTector SX100 instrument | LightEdge Technologies (Beijing, China) | | |
| Other | QuantaMaster 400 fluorimeter | Horiba (Kyoto, Japan) | | |

## Cell culture

ExpiCHO-S and Expi293F cell lines (Gibco, Thermo Fisher Scientific, Waltham, MA) were cultured in ExpiCHO Expression and Expi293 Expression media (Gibco, Thermo Fisher Scientific, Waltham, MA), respectively. Both cell lines were maintained at 37°C, 8% $CO_2$ with orbital shaking according to manufacturer's instructions. Expi293F cells were confirmed to be mycoplasma negative, while ExpiCHO-S were not tested.

## Antibodies

Monoclonal antibodies MAb362 isotypes $IgG_1$ and $IgA_1$ have been described before (*Ejemel et al., 2020*). REGN10987, S309, and CR3022 antibodies heavy and light variable region sequences (*Hansen et al., 2020*; *ter Meulen et al., 2006*; *Pinto et al., 2020*) were synthesized and cloned into pcDNA3.1 vector (Invitrogen, Thermo Fisher Scientific, Waltham, MA) in-frame with human IgG heavy or light chain Fc fragment. The recombinant constructs of heavy and light chain were transfected at 1:1 ratio into Expi293F cells using the ExpiFectamine 293 Transfection Kit (Gibco, Thermo Fisher Scientific, Waltham, MA). Four to five days after transfection the antibodies were purified from the supernatant by protein A affinity resin (ProSep-vA ultra, Millipore, Burlington, MA) and dialyzed into phosphate buffered saline pH 7.2 (PBS) overnight at 4°C. 2G12 mAb (*Buchacher et al., 1994*; *Scanlan et al., 2002*; *Trkola et al., 1996*), was expressed in ExpiCHO-S cells through co-transfection of plasmids encoding light and IgG heavy chains (Peter Kwong laboratory) using the ExpiFectamine CHO transfection kit (Gibco, Thermo Fisher Scientific, Waltham, MA) according to manufacturer's instructions. The antibody was purified from the cell culture supernatant 12 days post-transfection through protein G affinity resin (Thermo Fisher Scientific, Waltham, MA), and buffer exchanged and concentrated in PBS using centrifugal concentrators (Sartorius AG, Göttingen, Germany; Millipore, Burlington, MA). Monoclonal antibodies 4A8 and 1A9 were purchased from BioVision (Milpitas, CA) and GeneTex (Irvine, CA), respectively. Anti-6x-His-tag polyclonal antibody, and both horseradish peroxidase (HRP)-conjugated anti-mouse IgG Fc and anti-human IgG Fc were purchased from Invitrogen (Waltham, MA). Both HRP-conjugated anti-human kappa and anti-rabbit IgG were purchased from SouthernBiotech (Birmingham, AL) and Abcam (Cambridge, UK), respectively.

## Plasmids and site-directed mutagenesis

The mammalian codon-optimized gene coding SARS-CoV-2 (Wuhan-Hu-1 strain, GenBank ID: MN908947.3) glycoprotein ectodomain (SΔTM) (residues Q14–K1211) with SGAG substitution at the furin cleavage site (R682–R685), and proline substitutions at K986 and V987, was synthesized by GenScript (Piscataway, NJ) and inserted into pcDNA3.1(-). A C-terminal T4 fibritin foldon trimerization motif, a TEV protease cleavage site, and a His-tag were synthesized downstream of the SARS-CoV-2 SΔTM (*Figure 1B*). Insertion of A4 peptide (DSLDMLEW) at amino acid positions 161 and 345 in SARS-CoV-2 SΔTM was done through overlap-extension PCR (*Heckman and Pease, 2007*). The D614G amino acid change into both untagged and 161/345A4-tagged SΔTM constructs was done using the Q5 Site-Directed Mutagenesis Kit (NEB, Ipswich, MA) according to the manufacturer's instructions. Insertions and site-directed mutagenesis were confirmed through Sanger sequencing (GENEWIZ, Cambridge, MA).

## Protein expression and purification

Expression SΔTM trimers were performed by transfection of ExpiCHO-S cells with the plasmids described above using the ExpiFectamine CHO transfection kit (Gibco, Thermo Fisher Scientific, Waltham, MA) and according to manufacturer's instructions. SΔTM hetero-trimers for smFRET experiments were expressed by co-transfection with both the untagged SΔTM (D614 or D614G) construct and the corresponding 161/345A4-tagged SΔTM plasmid at a 2:1 molar ratio. Untagged SΔTM trimers or A4-tagged hetero-trimers were purified from cell culture supernatants as follows. Supernatants containing soluble SΔTM trimers were harvested 9 days post-transfection and adjusted to 20 mM imidazole, 1 mM $NiSO_4$, and pH 8.0 before binding to the Ni-NTA resin. The resin was washed, and protein was eluted from the column with 300 mM imidazole, 500 mM NaCl, 20 mM Tris-HCl pH 8.0, and 10% (v/v) glycerol. Elution fractions containing SΔTM were pooled and concentrated by centrifugal concentrators (Sartorius AG, Göttingen, Germany). The SΔTM protein was then further purified by size exclusion chromatography on a Superdex 200 Increase 10/300 GL column (GE

Healthcare, Chicago, IL) (*Figure 1—figure supplement 1*). Double 161/345A4-tagged SΔTM homo-trimers for functional assays were extracted from ExpiCHO-S cells at 6 days post-transfection with a non-denaturing lysis buffer (20 mM Tris-HCl pH 8.0, 500 mM NaCl, 10% (v/v) glycerol, 1% (v/v) Triton X-100, 2 mg/ml aprotinin, 1 mg/ml leupeptin, and 1 mg/ml pepstatin A (Sigma-Aldrich, St Louis, MO)). After 20 min of centrifugation at 4000 × *g*, the soluble fraction was diluted with two volumes of the same buffer without Triton X-100. These extracts were then passed through a 0.45 mm polyethersul-fone filter unit (Nalgene, Thermo Fisher Scientific, Waltham, MA), and the tagged SΔTM was purified by affinity chromatography using Ni-NTA agarose beads (Invitrogen, Waltham, MA) and size exclusion chromatography as described above.

A plasmid encoding soluble monomeric ACE2 with a C-terminal 6x-His-tag was transfected into ExpiCHO-S cells as described above. Supernatant containing ACE2 was harvested 6 days post-transfection, dialyzed at 4°C into 20 mM Tris-HCl pH 8.0, 500 mM NaCl, and 10% (v/v) glycerol buffer, using a 10 kDa MWCO dialysis membrane (Spectrum Repligen, Waltham, MA). For ACE2 purifica-tion, the dialyzed supernatant was supplemented with 20 mM imidazole pH 8.0 before purification as described above for SΔTM. Purified protein concentrations were estimated by UV absorbance at 280 nm and Bradford assay (Thermo Fisher Scientific, Waltham, MA). SΔTM concentration was also estimated by densitometric analysis of protein bands on immunoblots with the mAb 1A9 as described below, and using ImageJ software v1.52q (NIH, Bethesda, MD).

## PAGE and immunoblots

Protein expression was evaluated by denaturing PAGE in 4–20% acrylamide (Bio-Rad, Hercules, CA) and either staining with Coomassie blue or with immunoblots performed as follows. Protein gels were transferred into nitrocellulose membranes (Bio-Rad, Hercules, CA) according to the manufacturer's instructions. After 1 hr of blocking with 5% (w/v) skim milk in 0.1% (v/v) Tween-20 (Fisher Scien-tific, Hampton, NH) and PBS (PBS-T), membranes were incubated by shaking overnight at 4°C with dilutions 1:2000 in blocking buffer of the primary antibody. We used a rabbit anti-6x-His antibody (Invitrogen, Waltham, MA) to detect histidine-tagged proteins or mouse 1A9 antibody (GeneTex, Irvine, CA) for specific detection of SARS-CoV-2 SΔTM. Membranes were washed three times with PBS-T and then incubated with secondary HRP-conjugated anti-rabbit IgG (Abcam, Cambridge, UK) or anti-mouse IgG (Invitrogen, Waltham, MA) antibodies diluted in 0.5% (w/v) skim milk/PBS-T and incubated for 1 hr at room temperature. After three washes with PBS-T, membranes were developed using SuperSignal West Pico PLUS Chemiluminescent Substrate (Thermo Scientific, Waltham, MA) according to the manufacturer's instructions.

## ELISA

Ninety-six-well polystyrene plates (Thermo Scientific, Waltham, MA) were coated either with 200 ng of SARS-CoV-2 SΔTM proteins or bovine serum albumin (BSA, Thermo Scientific, Waltham, MA) through incubation overnight at 4°C. Plates were washed three times with PBS and blocked for 1 hr at room temperature with the immunoblot blocking buffer described above. After three washes with PBS, plates were incubated with 600 nM of the indicated antibodies diluted in PBS for 2 hr at room temperature. As secondary antibodies, HRP-conjugated anti-human kappa antibody (Souther-nBiotech, Birmingham, AL) diluted 1:4000 in PBS was used in wells treated with MAb362, CR3022, and S309 antibodies, while HRP-conjugated anti-human IgG Fc (Invitrogen, Waltham, MA) diluted 1:10,000 in PBS was used in wells treated with REGN10987, 4A8, and 2G12 antibodies. A 1:5000 dilu-tion of HRP-conjugated anti-mouse IgG Fc antibody in PBS was used in 1A9 antibody-treated wells. Plates were incubated with the secondary antibody dilutions for 1 hr at 37°C and developed with 1-Step Ultra TBM-ELISA (Thermo Scientific, Waltham, MA) reagent according to the manufacturer's instructions. The absorbances at 450 nm were measured using a Synergy H1 microplate reader (BioTek Winooski, VT). Absorbance values from non-specific binding to BSA-coated wells were subtracted from the values obtained for SΔTM-coated wells. The background-subtracted absorbance values were then normalized to the values obtained from antibodies binding to untagged SΔTM.

## Fluorescent labeling of proteins

Purified A4-tagged SΔTM hetero-trimers for smFRET imaging were prepared by overnight incubation at room temperature with 5 µM each of coenzyme A (CoA)-conjugated LD550 and LD650 fluorophores

(Lumidyne Technologies, New York, NY), 10 mM MgOAc, 50 mM HEPES pH 7.5, and 5 µM acyl carrier protein synthase (AcpS). Labeled protein was purified away from unbound dye and AcpS by size exclusion chromatography as above described, and elution fractions containing labeled SΔTM hetero-trimers were pooled and concentrated. Aliquots were stored at –80°C until use. Purified ACE2 was labeled with Cy5 conjugated to *n*-hydroxysuccinimide ester (Cytiva, Marlborough, MA) according to the manufacturer's instructions. ACE2 was then purified away from unbound dye by Ni-NTA affinity chromatography as described above, followed by buffer exchanged into PBS pH 7.4 using 10 kDa MWCO centrifuge concentrators (Millipore, Burlington, MA).

Purified LD550/LD650-labeled SΔTM spikes and Cy5-labeled ACE2 samples were analyzed by denaturing PAGE and in-gel fluorescence was visualized using a Typhoon 9410 variable mode imager (GE Amersham Biosciences, Amersham, UK) by laser excitation at 532 nm (emission filter: 580 BP 30 Cy3) to detect LD550, or 633 nm (emission filter: 670 BP 30 Cy5) to detect LD650 or Cy5 (*Figure 1—figure supplement 1*). To verify that changes in FRET could be interpreted as arising due to confor-mational transitions that reposition the fluorophores with respect to one another, bulk measurements of fluorescence lifetime and anisotropy were performed with a QuantaMaster 400 fluorimeter (Horiba, Kyoto, Japan). Measurements were performed in triplicate using 10 nM LD550/LD650-labeled SΔTM in the absence or presence of ACE2, under conditions identical to those described below for smFRET imaging, as well as using equivalent concentrations of unbound LD550 and LD650 fluorophores. While modest changes were seen when comparing free fluorophore to labeled SDTM, no significant changes in lifetime or anisotropy were seen when comparing unbound to ACE2-bound SΔTM.

## smFRET imaging and data analysis

Labeled SΔTM spikes (100 nM) were incubated in the absence or presence of 600 nM unlabeled ACE2 or the indicated antibody for 90 min at room temperature in Imaging Buffer (50 mM Tris-HCl pH 7.5, 50 mM KCl). This ligand concentration is approximately 100- to 1000-fold above reported dissociation constants for ACE2 and mAbs under consideration (*Chi et al., 2020*; *Ejemel et al., 2020*; *Hansen et al., 2020*; *Pinto et al., 2020*; *Walls et al., 2020*; *Yuan et al., 2020*; *Yurkovetskiy et al., 2020*). The exception here is mAb 2G12, which recognizes a glycan epitope with relatively low affinity (343 nM) (*Williams et al., 2021*). Our results may thus underestimate the effects of 2G12 on SΔTM confor-mation due to incomplete binding. Overall, reported rates of dissociation are in the range of $10^{-5}$ to $10^{-2}$ $s^{-1}$. Thus, it remains possible that after 90 min incubation, some of the SΔTM:ligand interactions may not be fully equilibrated. The 6x-His-tagged SΔTM was then immobilized on streptavidin-coated quartz microscope slides by way of Ni-NTA-biotin (vendor) and imaged using wide-field prism-based TIRF microscopy in Imaging Buffer as described (*Alsahafi et al., 2019*; *Das et al., 2020*; *Das et al.,*

**Table 2.** Förster resonance energy transfer (FRET)-state occupancies and rate constants for SΔTM D614.

Data are presented at mean ± standard error determined from the total population of traces analyzed.

| SARS-CoV-2 spikes D614 | FRET-state occupancies (%) | | Rate constants ($s^{-1}$) | |
|---|---|---|---|---|
| | Low-FRET (0.35) RBD-up conformation | High-FRET (0.65) RBD-down conformation | $k_{down}$ (Low to high FRET) | $k_{up}$ (High to low FRET) |
| Unbound | 40 ± 3 | 60 ± 3 | 3.8 ± 0.2 | 2.6 ± 0.2 |
| + ACE2 | 66 ± 3 | 34 ± 3 | 1.3 ± 0.1 | 2.2 ± 0.2 |
| + MAb362IgG₁ | 52 ± 2 | 48 ± 2 | 2.2 ± 0.1 | 2.2 ± 0.1 |
| + MAb362IgA₁ | 54 ± 2 | 46 ± 2 | 1.8 ± 0.1 | 2.0 ± 0.1 |
| + REGN10987 | 52 ± 3 | 48 ± 3 | 2.4 ± 0.2 | 2.2 ± 0.2 |
| + S309 | 49 ± 3 | 51 ± 3 | 2.3 ± 0.1 | 2.1 ± 0.1 |
| + CR3022 | 54 ± 4 | 46 ± 4 | 2.3 ± 0.2 | 3.1 ± 0.2 |
| + 4A8 | 48 ± 2 | 52 ± 2 | 2.3 ± 0.1 | 2.3 ± 0.1 |
| + 1A9 | 60 ± 2 | 40 ± 2 | 2.2 ± 0.1 | 3.4 ± 0.2 |
| + 2G12 | 59 ± 2 | 41 ± 2 | 2.5 ± 0.1 | 3.4 ± 0.2 |

**Table 3.** Förster resonance energy transfer (FRET)-state occupancies and rate constants for SΔTM D614G.

Data are presented at mean ± standard error determined from the total population of traces analyzed.

| | FRET-state occupancies (%) | | Rate constants (s⁻¹) | |
|---|---|---|---|---|
| SARS-CoV-2 spikes D614G | Low-FRET (0.35) RBD-up conformation | High-FRET (0.65) RBD-down conformation | $k_{down}$ (Low to high FRET) | $k_{up}$ (High to low FRET) |
| Unbound | 59 ± 3 | 41 ± 3 | 1.6 ± 0.2 | 2.0 ± 0.2 |
| + ACE2 | 69 ± 4 | 31 ± 4 | 0.7 ± 0.1 | 2.0 ± 0.4 |
| + MAb362IgG₁ | 58 ± 2 | 42 ± 2 | 2.0 ± 0.1 | 2.6 ± 0.1 |
| + MAb362IgA₁ | 63 ± 2 | 37 ± 2 | 1.6 ± 0.1 | 2.6 ± 0.1 |
| + REGN10987 | 63 ± 4 | 37 ± 4 | 2.0 ± 0.2 | 3.7 ± 0.3 |
| + S309 | 54 ± 3 | 46 ± 3 | 2.2 ± 0.1 | 2.4 ± 0.2 |
| + CR3022 | 65 ± 4 | 35 ± 4 | 1.1 ± 0.2 | 1.8 ± 0.2 |
| + 4A8 | 57 ± 3 | 43 ± 3 | 2.0 ± 0.2 | 2.3 ± 0.2 |
| + 1A9 | 66 ± 3 | 34 ± 3 | 1.7 ± 0.1 | 3.2 ± 0.2 |
| + 2G12 | 69 ± 3 | 31 ± 3 | 1.4 ± 0.1 | 2.7 ± 0.3 |

*2018*; *Durham et al., 2020*). Imaging was performed in the continued presence of ligands at room temperature and smFRET data were collected using Micromanager (*Edelstein et al., 2014*) v2.0 ( micro-manager.org) at 25 frames/s. All smFRET data were processed and analyzed using the SPARTAN software (https://www.scottcblanchardlab.com/software) in Matlab (Mathworks, Natick, MA) (*Juette et al., 2016*). Figures displaying smFRET data were also prepared in Matlab. smFRET traces were identified according to following criteria: mean fluorescence intensity from both donor and acceptor were greater than 50, duration of smFRET trajectory exceeded five frames, correlation coefficient calculated from the donor and acceptor fluorescence traces ranged between –1.1 and 0.5, and signal-to-noise ratio was greater than 8. Traces that fulfilled these criteria were then verified manually. Traces from each of three technical replicates were then compiled into FRET histograms and the mean probability per histogram bin ± standard error were calculated, as shown in *Figure 3* and *Figure 4—figure supplement 1*. Traces were idealized to a three-state HMM (two non-zero-FRET states and a 0-FRET state) and the transition rates were optimized using the maximum point likelihood (MPL) algorithm (*Qin et al., 2000*) implemented in SPARTAN. The rates reported in *Figures 3 and 4*, and *Tables 2 and 3* reflect the mean ± standard error determined from three technical replicates. The three-state model was selected by comparing the Akaike information criterion (AIC) across multiple different models with a range of state numbers and topologies. For each model the maximized log-likelihood per trace was estimated using the MPL algorithm. The AIC values were calculated according to $AIC_i = 2 \cdot N_i - 2 \cdot LL_i$, where $N_i$ and $LL_i$ are the number of model parameters and the maximized log-likelihood per trace for the $i$th model considered. In this analysis, of the models considered, the model with the minimum AIC value is taken to be the model that best reflects the experimental data. Accordingly, we found that the three-state model adequately reflects the data, with no further reduction in AIC resulting from inclusion of addition states or model parameters (*Figure 3—figure supplement 1*). The idealizations from the total population of traces analyzed were used to construct Gaussian distributions, which were overlaid on the FRET histograms to visualize the results of the HMM analysis. These idealizations were also used to calculate the distribution in occupancies (fraction of time until photobleaching) in the 0.65- and 0.35-FRET states across the total population of traces. The distributions in occupancies were used to construct violin plots in Matlab, as well as calculate median occupancy, mean occupancy, 25th and 75th quantiles, and standard errors, as displayed in *Figures 3 and 4*, and *Table 2*; *Table 3*. Statistical significance measures (p-values) were determined by one-way ANOVA in Matlab. This analysis displays the full breadth of dynamic behavior across the total population of traces analyzed. The total number of traces analyzed was sufficient to ensure minimally 85% statistical power during comparison of occupancy data from unbound to ligand-bound SΔTM.

## FCS-based ACE2-binding assay

ACE2 binding to the untagged and A4-tagged SΔTM spikes was evaluated by FCS as follows. Several concentrations ranging from 0.1 to 200 nM SΔTM were incubated with 100 nM Cy5-labeled ACE2 in PBS pH 7.4 for 1 hr at room temperature. Where indicated, 200 nM SΔTM was incubated with 600 nM of the indicated antibody for 1 hr at room temperature, before adding 100 nM Cy5-labeled ACE2. Non-specific antibody binding to Cy5-labeled ACE2 was determined by incubation in the absence of SΔTM. Samples were then placed on No. 1.5 coverslips (ThorLabs, Newton, NJ) and mounted on a CorTector SX100 instrument (LightEdge Technologies, Beijing, China) equipped with a 638 nm laser; 10–25 autocorrelation measurements per independent experiment were made for 10 s each at room temperature for each condition. To obtain the fractions of unbound and bound ($f$) ACE2 after incubation with SΔTM, normalized autocorrelation curves were fit to a model of the diffusion of two species in a three-dimensional Gaussian confocal volume (*Ducas and Rhoades, 2012*; *Ries and Schwille, 2012*),

$$G(\tau) = (1 - f) \cdot g_{\text{unbound}}(\tau) + f \cdot g_{\text{bound}}(\tau),$$

where

$$g_i(\tau) = \left(1 + \frac{\tau}{\tau_i}\right)^{-1} \left(1 + \frac{\tau}{s^2 \tau_i}\right)^{-1/2}$$

and $t_i$ is the diffusion time for bound or unbound ACE2 and $s$ is the structure factor that parameterizes the dimensions of the confocal volume. To determine $t_{\text{unbound}}$ FCS data was obtained for ACE2 in the absence of SΔTM and fit to a model of a single diffusing species ($f = 0$). This value was then fixed during fitting of the FCS data obtained after incubation of ACE2 with SΔTM, so that only $t_{\text{bound}}$ and $f$ were allowed to vary. ACE2 binding was expressed as the average bound fraction ($f$) at each SΔTM concentration normalized either to the fraction bound at the highest SΔTM concentration (*Figure 2C*) or to the fraction bound in the absence of antibodies (*Figure 5C*). All fitting was performed with a non-linear least-squares algorithm in MATLAB (The MathWorks, Waltham, MA). Dissociation constants ($K_D$) were determined using GraphPad Prism version 9.2.0 (GraphPad Software, San Diego, CA).

## Structural analysis

Protein structures from RCSB PDB were visualized and analyzed using PyMOL software version 2.0.7 (The PyMOL Molecular Graphic System, Schrödinger Inc, New York, NY).

## Correlation and statistical analysis

Data sets subjected to statistical analysis were first tested for normality using GraphPad Prism version 9.2.0 (GraphPad Software, San Diego, CA). Where indicated, statistical significances were evaluated through either two-tailed parametric (unpaired t-test with Welch's correction) or nonparametric (unpaired Mann-Whitney) tests. Both tests were performed with 95% confidence levels and p-values < 0.05 were considered significant. Significance values are indicated as *p < 0.05, **p < 0.01, ***p < 0.001, ****p < 0.0001. Two-tailed nonparametric Spearman test with 95% confidence was performed to evaluate the correlation level between the occupancy of SΔTM in the open conformation due to allosteric antibody binding and ACE2 binding (*Figures 4 and 5*). The correlation level between the above variables was determined according to established criteria (*Schober et al., 2018*) regarding Spearman coefficients ($r_s$) rank values as follows: 0.00–0.10 = 'negligible', 0.10–0.39 = 'weak', 0.40–0.69 = 'moderate', 0.70–0.89 = 'strong', and 0.90–1.00 = 'very strong' correlation.

## Acknowledgements

The authors thank Dr Natasha Durham and Dr Aastha Jain (UMass Chan Medical School) for critical discussion and reading of the manuscript, as well as Dr Peter Kwong (NIAID, NIH) for kindly providing molecular clones of the 2G12 antibody heavy and light chains. Funding: This work was supported by UMass Chan Medical School COVID-19 Pandemic Relief Fund (JBM), National Institutes of Health grants R37AI147868 (JL), R01AI148784 (JL, JBM), and K22CA241362 (KS), Evergrande COVID-19 Response Fund (JL), Massachusetts Consortium on Pathogen Readiness (JL), and Worcester Foundation for Biomedical Research (KS).

## Additional information

### Competing interests

Qi Li, Monir Ejemel, Yang Wang: A patent application has been filed on May 5, 2020 on monoclonal antibodies targeting SARS-CoV-2 U.S. Patent and Trademark Office patent application no. 63/020,483; patent applicants: YW, ME, and QL. The other authors declare that no competing interests exist.

### Funding

| Funder | Grant reference number | Author |
|---|---|---|
| UMass Chan Medical School COVID-19 Pandemic Relief Fund | | James B Munro |
| National Institutes of Health | R37AI147868 | Jeremy Luban |
| National Institutes of Health | R01AI148784 | Jeremy Luban James B Munro |
| National Institutes of Health | K22CA241362 | Kuang Shen |
| Evergrande COVID-19 Response Fund | | Jeremy Luban |
| Massachusetts Consortium on Pathogen Readiness | | Jeremy Luban |
| Worcester Foundation for Biomedical Research | | Kuang Shen |

The funders had no role in study design, data collection and interpretation, or the decision to submit the work for publication.

### Author contributions

Marco A Díaz-Salinas, Data curation, Formal analysis, Investigation, Writing – original draft, Writing – review and editing, Methodology, Visualization; Qi Li, Monir Ejemel, Leonid Yurkovetskiy, Resources; Jeremy Luban, Kuang Shen, Yang Wang, Supervision, Writing – review and editing; James B Munro, Conceptualization, Data curation, Formal analysis, Funding acquisition, Supervision, Writing – original draft, Writing – review and editing

### Author ORCIDs

Marco A Díaz-Salinas (iD) http://orcid.org/0000-0003-2983-0123
Jeremy Luban (iD) http://orcid.org/0000-0001-5650-4054
James B Munro (iD) http://orcid.org/0000-0001-7634-4633

### Decision letter and Author response

Decision letter https://doi.org/10.7554/eLife.75433.sa1
Author response https://doi.org/10.7554/eLife.75433.sa2

## Additional files

### Supplementary files
• Transparent reporting form

### Data availability
All data generated or analyzed during this study are included in the manuscript and supporting files.

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
