## [Editor Report]

Using single-molecule fluorescence imaging, the authors of this paper characterize a conformational change of the Spike protein from severe acute respiratory syndrome coronavirus 2 (SARS-CoV-2) that is important for the ability of the Spike protein to target its receptor on the surface of a host cell and facilitate viral entry into the cell. The authors characterize how interactions of the Spike protein with its receptor and with Spike protein-targeting neutralizing antibodies alter this conformational change. The results provide important insights into the mechanisms-of-action of several classes of neutralizing antibodies.

---

## [Decision Letter]

**Decision letter after peer review:**

Thank you for submitting your article "Conformational dynamics and allosteric modulation of the SARS-CoV-2 spike" for consideration by *eLife*. Your article has been reviewed by 2 peer reviewers, one of whom is a member of our Board of Reviewing Editors, and the evaluation has been overseen by Richard Aldrich as the Senior Editor. The following individual involved in the review of your submission has agreed to reveal their identity: Mark E Bowen (Reviewer #2).

Essential revisions:

1) Please revise your manuscript to describe your model selection process more comprehensively and, most importantly, to include a stronger justification for your use of a two-state model.

2) Please revise your manuscript to include a stronger justification for the incubation times and temperatures that were used to achieve equilibration/saturation of ligand binding.

3) Please revise your manuscript to include a stronger justification for the relatively small number of smFRET trajectories recorded and analyzed in these studies. Related, please clarify the number of trajectories in Figure 3 and Figure 4-Supp 1, as requested b Reviewer #1.

4) Please revise your manuscript to carefully and comprehensively address the error analysis concerns raised by Reviewer #1.

*Reviewer #1 (Recommendations for the authors):*

1. When analyzing the smFRET traces, how did the authors arrive at the conclusion that a two-state conformational model was the most appropriate model for the analysis? In the Methods, the authors actually describe using a three-state model, but I think only two of those states represent distinct conformations of the S protein (up and down) whereas one of those states is a "zero" FRET state presumably arising from photobleaching. What metric did the authors use to determine that a two-state conformational model was most appropriate? How did that metric change when considering a three-state conformational model, a four-state conformational model, etc.?

2. In the Methods section, the authors should describe how they tested and/or determined that 60-90 min, room temperature incubation times were enough for equilibration/saturation of ligand binding.

3. The number of smFRET traces obtained and analyzed at each experimental condition is lower than what is reported in typical smFRET studies. The authors should explicitly state why this is and what limits the number of traces they can collect. Also, in the histograms shown in Figure 3, it is not clear whether the "N" that is reported for each histogram is the total number of traces for all "three independent populations of smFRET traces" or the number of traces for each of the three "independent populations". This should be clearly specified in Figure 3. Moreover, in Figure 4-Supp 1, the "N"s are not reported at all; they should be reported and it should also be specified whether they report the total number of traces in all three "independent populations" or the number of traces for each of the three "independent population".

4. In Figure 3 and Figure 4-Supp 1, what is meant by "three independent populations of smFRET traces"? The authors should specify whether the smFRET experiments were independently repeated and, if so, how many times they were repeated and whether these repetitions were technical replicates or biological replicates. If the experiment was not independently repeated, then this should be explicitly stated and the authors should provide some explanation or justification for not repeating the experiments. Related to these issues, the error bars on the FRET histograms report the means and standard deviations of the "three independent populations of smFRET traces" whereas the error bars on the FRET-based occupancies report the means and standard deviations "across the population of traces" and the error bars on the FRET-based rates were "estimated from 1000 boot-strapped samples". It is not clear why the error analyses have been performed in all of these different ways for the different parameters extracted from the FRET data. I imagine that the largest source of error in these experiments is the ability of the researcher to repeat the experiment. If so, shouldn't the error bars on all of these parameters report the mean and standard deviation obtained from three independently measured biological replicates? The authors should clearly describe and carefully justify their error analyses, particularly given the relatively small effects that are reported for some of the parameters.

5. I think the use of "high FRET" and "low FRET" throughout the manuscript is distracting in that I need to constantly remind myself that these FRET states have been assigned to the "down" and "up" conformational states. The authors seem to be at least somewhat aware of this potential issue in that they periodically explicitly remind the reader of these assignments. I think the manuscript would flow a lot better if, once the "high" and "low" FRET states are explicitly assigned to the "down" and "up" conformational for the first time on p4, the terms "down" and "up" are used throughout the rest of the manuscript, figures, figure captions, etc.

*Reviewer #2 (Recommendations for the authors):*

I would favor more discussion of the experimental outcomes relative to Lu et al. since this is in press. Both use smFRET to investigate conformation and dynamics upon ACE and Ab binding. Are the conclusions, regarding the number of states and whether ACE actively shifts the conformational ensemble, at odds?

The changes in UP/DOWN transition rates in the presence of ACE or Ab contains a contribution from the binding kinetics of the partner to the Spike, which is not presented. How do these timescales compare? Does proteins stay bound to the spike for the duration of the state occupancy or mediate its effect by binding and departing before the state switch. Are intramolecular FRET possible using the labeled construct from FCS studies with surface attached spike proteins to get these kinetics?

The changes in transition rate constants and ACE binding from antibodies are small being mostly within a factor of 2 or 3. How is the efficacy of the antibodies or triggering of fusion reconciled with the moderate effects on the protein conformation?

The work is interpreting FRET changes as indicating underlying changes in molecular distance. There are not any photophysical measurements of dye properties under the different conditions to rule out photophysical origins of these effects (specifically, Protein Induced Fluorescence Enhancement from bound proteins).

It is not clear what is being concluded based on the transition density plots. Is it the overall number of transitions that is changing? Are all molecules dynamic?

The figures could use a better balance between graphical elements and blank space. The panels are small within a large white field, which makes some of the panels hard to read.

The structural element in Figure 2 could be smaller to allow the data to be the focus.

---

## [Author Response]

Essential revisions:1) Please revise your manuscript to describe your model selection process more comprehensively and, most importantly, to include a stronger justification for your use of a two-state model.

We appreciate the importance of the Reviewers’ comment and agree that a discussion of model selection is appropriate. In brief, we used the Akaike information criterion (AIC) to compare maximized likelihoods across a range of models with varying numbers of state and topologies. The simplest model we considered was a two-state model, consisting of one non-zero-FRET state and a 0-FRET state, which accounts for fluorophore photobleaching. We find that addition of a second non-zero FRET state (three states in total) improves model fitness as evidenced by a decrease in the AIC value. Additional states and connections did not further decrease the AIC value for the SDTM D614 or D614G data sets. Therefore, we performed our analyses using a circular model with 3 FRET states in total (0.65, 0.35, and 0 FRET). This analysis is now described in the Materials and methods section (lines 507-516 of the revised document) and presented in Figure 3—supplement 1.

2) Please revise your manuscript to include a stronger justification for the incubation times and temperatures that were used to achieve equilibration/saturation of ligand binding.

The Reviewers raise an important point, which is often overlooked. Spike proteins (S trimers and isolated domains) have been used in multiple studies to characterize the interaction with ACE2 and antibodies. However, these studies have yielded divergent estimates of ligand affinity for the spike proteins, likely stemming from the different protein constructs and reaction conditions used. For the ligands studied here, estimates of K_D_ range from 0.04-8 nM. Overall, we have sought to work in high excess of these K_D_ values. The outlier is 2G12, which binds a glycan epitope with relatively low affinity (343 nM). Therefore, our studies were likely conducted under sub-saturating 2G12 binding. So, our results probably under-estimate the effects of 2G12 on S conformation and ACE2 binding. Regarding equilibration, the rates of dissociation have not been thoroughly reported for the ligands considered here. The available rates are in the range of 10^-5^ to 10^-2^ s-1. So, some of our measurements (which followed a 90-min incubation) may not have been conducted under fully equilibrated conditions. These caveats are now noted in the Materials and methods section on lines 479-488.

3) Please revise your manuscript to include a stronger justification for the relatively small number of smFRET trajectories recorded and analyzed in these studies. Related, please clarify the number of trajectories in Figure 3 and Figure 4-Supp 1, as requested b Reviewer #1.

Here again, we thank the reviewers for raising an important question that often goes overlooked in smFRET studies. To the best of our knowledge, there is no consensus on the minimum number of traces needed to justify a conclusion. In the present study we have chosen to determine the adequate number of traces by calculating the sample size necessary to obtain the desired statistical power for a given hypothesis. In this case, our hypothesis is that addition of the specified ligand shifts the occupancy in the RBD-up conformation. As seen in Author response image 1, for the magnitude of the change in RBD-up occupancy that we observed upon addition of ACE2 to SDTM, and the observed variance across the population of traces analyzed, a sample size of 140 traces yields a statistical power of >99%. This indicates that the observed change in the RBD-up occupancy upon addition of ACE2, and the associated p-value, are highly reliable. In the second example shown in Author response image 1, for the effect on RBD-up occupancy seen upon addition of mAb 4A8, which is a comparatively modest effect, our analysis of 294 traces yields a statistical power of approximately 90%. All data sets displayed in Author response image 1 achieve minimally 85% statistical power. Again, this indicates that our results and the associated p-values are highly reliable. We are therefore confident that the number of traces that we have analyzed is sufficient to justify the conclusions of our study. This is now noted in the Materials and methods section on lines 524-526.

**Author response image 1. sa2fig1:** Statistical power achieved for a given number of traces. Data for MAb362-IgG and MAb362-IgA overlap with REGN10987 and CR3022, respectively.

4) Please revise your manuscript to carefully and comprehensively address the error analysis concerns raised by Reviewer #1.

We appreciate the Reviewers raising this issue, and we fully acknowledge that the treatment of our smFRET data was insufficiently described in the initial submission. To address this concern, we have expanded the statistical analysis of the FRET state occupancy data, and thoroughly described our procedures in the Materials and methods, and in the legend to Figure 3. In brief, we have clarified the origins of the reported error bars. The errors bars in the FRET histogram bin counts reflect standard errors, which were determined from three technical replicates, as Reviewer 1 surmised, but which we had not stated clearly. In the previous submission, the error bars reported with the rate constants had reflected 95% confidence intervals generated by bootstrapping in Matlab. However, to simplify the interpretation of our data, in the revised submission the rate constant error bars now reflect standard errors determined across the three technical replicates. Although the recalculated error bars are indistinguishable from the error bars in the initial submission, the internally consistent procedure makes for a more logical description of our methods. For this reason, we appreciate the rigorous comments from the Reviewer 1. These procedures are now reported in the Materials and methods, in the subsection entitled “smFRET imaging and data analysis”.

Finally, in order to transparently display the breadth in the behavior of the smFRET traces that we have observed, we have now reported the FRET state occupancy data as violin plots from the total population of traces analyzed. This treatment enabled us to calculate mean, median, and quantiles in FRET-state occupancies across the experimental conditions considered. This recalculation in mean occupancies and standard errors reported in Tables 2 and 3 led to adjustments from the initial submission, but the qualitative conclusions still hold. This treatment also enabled us to calculate statistical significance measures (p-values) using ANOVAs. As Reviewer 1 correctly points out, some of the effects that we report are relatively modest in magnitude. We therefore felt it was important to report p-values in this way. This procedure is now described in the Materials and methods, lines 516-524.